# Comparative Transcriptomic Analysis Reveals the Functionally Segmented Intestine in Tunicate Ascidian

**DOI:** 10.3390/ijms24076270

**Published:** 2023-03-27

**Authors:** Wei Zhang, An Jiang, Haiyan Yu, Bo Dong

**Affiliations:** 1Fang Zongxi Center, MoE Key Laboratory of Marine Genetics and Breeding, College of Marine Life Sciences, Ocean University of China, Qingdao 266003, China; 2Laoshan Laboratory for Marine Science and Technology, Qingdao 266237, China; 3MoE Key Laboratory of Evolution and Marine Biodiversity, Ocean University of China, Qingdao 266003, China

**Keywords:** *Ciona*, intestine, transcriptome, segmentation

## Abstract

The vertebrate intestinal system consists of separate segments that remarkably differ in morphology and function. However, the origin of intestinal segmentation remains unclear. In this study, we investigated the segmentation of the intestine in a tunicate ascidian species, *Ciona savignyi,* by performing RNA sequencing. The gene expression profiles showed that the whole intestine was separated into three segments. Digestion, ion transport and signal transduction, and immune-related pathway genes were enriched in the proximal, middle, and distal parts of the intestine, respectively, implying that digestion, absorption, and immune function appear to be regional specializations in the ascidian intestine. We further performed a multi−species comparison analysis and found that the *Ciona* intestine showed a similar gene expression pattern to vertebrates, indicating tunicates and vertebrates might share the conserved intestinal functions. Intriguingly, vertebrate pancreatic homologous genes were expressed in the digestive segment of the *Ciona* intestine, suggesting that the proximal intestine might play the part of pancreatic functions in *C. savignyi*. Our results demonstrate that the tunicate intestine can be functionally separated into three distinct segments, which are comparable to the corresponding regions of the vertebrate intestinal system, offering insights into the functional evolution of the digestive system in chordates.

## 1. Introduction

The vertebrate digestive system is composed of the digestive tract and digestive glands. The digestive tract, which is a long tube varying in diameter, consists of different regions, such as the mouth, pharynx, esophagus, stomach, small intestine, and large intestine [1]. The intestine, as a complex coordinated organ, has multiple physiological functions, including digestion, nutrient absorption, and endocrine secretion [2]. The intestine is also considered to play a regulatory role in food intake, insulin secretion, and energy balance in the metabolism of the organism [3]. Additionally, the intestine is also the largest “immune organ” of the human body, maintaining the stability of the intestinal environment and the normal physiological activities of the human body through interaction with intestinal microbiota [4,5,6]. The intestine of vertebrates is subdivided into many regions, including the large intestine, small intestine, colon, and rectum [1,7,8,9]. However, in basal vertebrates, such as lamprey, the rectum, segmented intestine, and salivary glands cannot be identified morphologically [9,10].

The origin of intestinal segmentation remains elusive. Molecular features associated with different intestine segments provide a potential approach to this question. Ascidians, as the closest living relatives of vertebrates [11,12], have a digestive tube, including an oral siphon, pharynx, esophagus, stomach, and intestine [13]. The ascidian intestine is a tubular structure closely connected to the stomach. Unlike the specialized intestinal zones of higher vertebrates, the morphology of the ascidian intestine is relatively simple. The segmentation of the ascidian intestine has already been described at both morphological and molecular levels based on the expression of several marker genes [13,14,15]. The intestine transcriptome of *C. robusta* has been reported, and the intestine−specific genes were screened and compared with other organs from different species [16]. However, in the case of *C. savignyi*, another species of the *Ciona* genus, gene analysis on the intestine at the transcriptome level has not been performed yet. As the closest relatives to vertebrates, the evolutionary relationship between ascidian intestinal segmentation and the digestive organs in vertebrates is unclear.

In this study, we carried out transcriptome sequencing of the whole intestine of the ascidian *C. savignyi* to build and analyze its expression profile. Then, comparative transcriptome analyses were performed to find out the functional relationship between the intestine and the stomach of *C. savignyi*. Subsequently, expressions of the orthologs of vertebrate digestive organ-specific genes in the *C. savignyi* intestine were analyzed to search for the connectivity between the intestine of *C. savignyi* and the digestive system of vertebrates. Ultimately, we also performed a multi−species comparison of the conservation and divergence of intestine function during evolution. 

## 2. Results

### 2.1. Gene Enrichment Analysis Reveals the Segmented Expression Patterns of C. savignyi Intestine

To facilitate sample collection, the ascidian intestines were divided into three segments equally, including proximal, middle, and distal fragments (Figure 1A). The sequenced data are displayed in Appendix A.

We first examined the Hox gene expression in the intestine, and the results showed that *Ci-Hox2*, *Ci-Hox4*, and *Ci-Hox5/6* were expressed at the proximal intestine, *Ci-Hox10* was expressed in the middle intestine, and *Ci-Hox1*, *Ci-Hox3*, and *Ci-Hox12* were expressed in the distal intestine (Figure 1B). The regional expression of Hox genes along the anterior–posterior axis indicated the heterogenization of the intestine in *C. savignyi*.

Next, we performed a function enrichment analysis on all expression genes (FPKM > 1). The results of the GO enrichment analyses showed that the enriched terms of the expressed genes in three segments were similar in Biological Process (BP), Cellular Component (CC), and Molecular Function (MF). However, genes in the proximal segment were significantly enriched in catabolic process, peptide metabolic process, organic substance catabolic process, and other catabolic processes (padj < 0.05) (Appendix A), indicating the stronger digestive function potential in the proximal segment. The genes in the middle were significantly enriched in the terms associated with organic substance transport and establishment of protein localization genes (padj < 0.05) (Appendix A), indicating a stronger absorption function in the middle segment. In the distal segment, most of genes were mainly significantly enriched in Small GTPase mediated signal transduction, demonstrating the establishment of localization in cell and intracellular transport (padj < 0.05) (Appendix A).

To systematically investigate the expression changes of genes with biologically similar functions in the intestine, we performed a trend expression analysis to identify co-expression modules reflecting functional responses of biological relevance. A total of eight clusters were divided (Figure 1C), in which four clusters were statistically significant (*p*-value < 0.05) (Figure 1C). The expression levels of genes in cluster 0 were down-regulated, whereas the expression levels of genes in cluster 7 were up-regulated gradually along the proximal, middle, and distal segments (Figure 1D,G). The expression trend of genes in cluster 3 was down-regulated and higher in the proximal and middle relative to the distal segment (Figure 1E). In cluster 4, gene expression was up-regulated and lower in the proximal and middle compared with the distal segment (Figure 1F). The trend of gene expression of clusters 4 and 3 indicates that the proximal and middle segments share parts of similar functions.

To characterize these four clusters individually, a GO enrichment analysis of genes was conducted for each cluster. The genes in cluster 0 were enriched in digestive enzyme and catabolic process-related terms, e.g., macromolecule catabolic process, protein catabolic process, peptide metabolic process, catabolic process, purine nucleoside monophosphate and ATP metabolic/biosynthetic process, and organic substance catabolic process (Figure 1D and Appendix A), which further suggests that the proximal segment possibly has a digestive function. The genes in cluster 3 were associated with macromolecule biosynthetic and nucleic acid metabolic process, response to stress, and DNA replication and DNA damage stimulus (Figure 1E and Appendix A). The terms of signal transduction, signal transducer/receptor activity, and G-protein coupled receptor activity were identified in cluster 4 (Figure 1F and Appendix A). Cluster 7 was associated with myosin complex, Rho GTPase binding, tumor necrosis factor receptor superfamily binding, and tumor necrosis factor receptor binding (Figure 1G and Appendix A). Rho GTPases are actin regulators that are involved in various biological processes including immune activities. These results suggest that the distal segment plays a potential immune function. KEGG enrichment analysis showed the genes in cluster 0 and 3 were enriched in oxidative phosphorylation, ribosome, proteasome, DNA replication, and so on, while cluster 4 and 7 were enriched in ECM-receptor interaction and Glycosaminoglycan degradation (*p* < 0.05) (Appendix A). 

To further investigate the functional difference among the different segments of the intestine, DEGs were selected and analyzed. By comparing the proximal segment with the middle one, we found that 184 genes were up-regulated and 222 genes were down-regulated (Figure 2A and Appendix A). *Cap2*,* PNLIP*,* Acp5*, and *Trypsin* showed significant up-regulation and *FCGBP* showed the opposite trend (Appendix A). Among the total of 4178 DEGs, 2424 genes were up-regulated and 1754 genes were down-regulated by comparing the proximal segment to the distal segment (Figure 2A and Appendix A). The expression of genes *TNF*,* hmcn1*,* slc26c*,* EVX*,* Foxq*,* pax,* and *Muc* was opposite to that of *Cyp1b1* and *creatine kinase*, which were significantly up-regulated (Figure 2A and Appendix A). Compared with the distal segment, 1496 up-regulated genes and 705 down-regulated genes were screened in the middle segment, among which genes *cpg1*, *TNF*, and *slcF9b2* were significantly down-regulated (Figure 2A and Appendix A). The relatively small amount of DEGs between the distal and middle segments suggests that the two segments potentially share similar functions. 

To characterize three intestine segments, a functional enrichment analysis of DEGs was conducted (Figure 2B and Appendix A; Figure 2C and Appendix A; Figure 2D and Appendix A). Compared with the middle and distal segments, the up-regulated genes in the proximal segment were enriched in the peptide metabolic process, peptide biosynthetic process, amide biosynthetic process, aminoglycan metabolic process, and other digestive and metabolic processes. Compared with the proximal and distal segments, small molecule metabolic process, ATP metabolic process, ATP biosynthetic process, and some ion binding activities were enhanced in the middle segment. In the distal segment, immune system process, response to oxidative stress, oxidoreductase activity, and peroxidase activity were enriched compared with the proximal and middle segments. The qPCR results validated the reliability of the transcriptome data on the tissue−specific−expressed genes in three segments (Appendix A). In general, enrichment analysis of DEGs indicated the potential digestion, absorption, and immune functions of the proximal, middle, and distal segments, respectively.

We then screened the tissue−specific−expressed genes from all the expressed genes in each intestine segment to further probe the functional heterogenization of the intestine in *Ciona*. As shown in the Venn diagram, 369, 299, and 711 genes were specifically expressed in the proximal, middle, and distal segments, respectively (Figure 2E). GO enrichment analysis showed that the tissue-specific-expressed genes in the proximal and middle segments presented similar expression patterns and focused on G-protein coupled receptor signaling pathway, signaling receptor activity, molecular transducer activity, and transmembrane signaling receptor activity (Figure 2E,F). However, the tissue-specific-expressed genes in the middle segment were specifically enriched in transmembrane transporter activity and anion transmembrane transporter activity, related to absorption, indicating that the middle part may have specific absorptive functions. The tissue−specific−expressed genes in the distal segment were related to glycosylation, glycoprotein metabolic/biosynthetic process, transferase activity, and binding terms (Figure 2F), which play crucial roles in immunity.

Taken together, the differential expression of Hox genes among segments and the GO enrichment results of all genes suggest the functional differentiation of *Ciona* intestine. Expression tendency analysis showed that the genes in significantly different clusters were enriched in pathways related to digestion, absorption, and immunity, indicating that these genes have a trend of changes in the functional segment. These results suggest that the proximal and middle segments possibly share a similar function. Additionally, enrichment analyses of all DEGs and tissue-specific-expressed genes showed that digestion, absorption, and immune functions were possibly dominant in the proximal, middle, and distal segments, respectively. Therefore, the *Ciona* intestine could be divided into digestion, absorption, and immune segments based on expression profiles of all genes, specific expression of Hox genes, and enrichment analyses of all DEGs and tissue-specific-expressed genes (Figure 1H).

### 2.2. The Functional Relationship between Intestine and Stomach 

The stomach and intestine are the main organs in the digestive system. The ascidian intestine is connected to the stomach (Figure 1A). To unravel the functional relationship between the intestine and stomach, a comparative analysis was performed between the different segments of the intestine and stomach. In total, 473, 728, and 4886 DEGs were found in the proximal, middle, and immune segments of the intestine, respectively, compared with the stomach (Figure 3A). The heat map of DEGs showed that the stomach and the proximal digestion segment of the intestine share higher similarity than the other two segments (Appendix A). However, our results showed the significant differences between the intestine and stomach, with a gradually increasing trend. More DEGs were detected in the distal immune segment versus the stomach, suggesting that the distal immune segment was functionally different from the stomach. The results of KEGG enrichment (Figure 3B) and GO enrichment analysis (Figure 3C–E) supported this conclusion as well (Appendix A).

However, GO enrichment analysis showed that almost all of the terms were for more gene enrichment in the stomach rather than in the intestine. Compared with all segments of the intestine, the up-regulated genes of the stomach were related to cytoskeleton and metabolic process pathways involving peptidase activity, endopeptidase activity, and L-amino acid peptides, indicating that the stomach has a relatively higher digestive function (Figure 3C–E). There were more ion transport-related pathways (e.g., ion transport, anion transport, and copper ion binding) and transmembrane transporter activity in the absorption segment, indicating that the intestine and stomach have a similar absorption function, but the stomach is slightly stronger (Figure 3D). Compared to the immune segment, more genes of the stomach were enriched in the energy synthesis-related pathway, including ATP biosynthesis process, purine nucleoside triphosphate, ribonucleoside triphosphate, and purine ribonucleoside triphosphate, indicating that the stomach probably possesses higher ATP biosynthesis capacity (Figure 3E).

### 2.3. Expression of the Orthologs of Vertebrate Digestive Organ−Specific Genes in the C. savignyi Intestine

The ascidian adult has a relatively simple digestive system compared to vertebrates. To investigate whether the *C. savignyi* intestine has a similar gland function to vertebrates’, we compared the gene expression of the *C. savignyi* intestine to that in a variety of vertebrate digestive organs. First, we linked the expression of *Ciona* intestinal genes with *M. musculus* digestive organs. The result showed that all the genes found in the transcriptome data of *C. savignyi* intestine were detected in the large intestine, small intestine, pancreas, and liver of *M. musculus* (Figure 4A and Appendix A). Next, we confirmed the counts of orthologous genes between the intestine of *C. savignyi* and duodenum, small intestine, large intestine (colon), pancreas, and liver of *H. sapiens*, respectively (Figure 4B and Appendix A). The results obtained from the comparison against *H. sapiens* were similar to those in *M. musculus*. Importantly, the number of orthologous genes from the digestion segment accounts for the largest proportion in the small intestine and pancreas of *M. musculus* and *H. sapiens*. The number of orthologous genes from the immune segment is the largest proportion in the large intestine. Based on these results, it seems that the *C. savignyi* digestion segment may share conservative functions with the small intestine and pancreas in vertebrates, and that the *C. savignyi* immune segment has a functional similarity to the large intestine of vertebrates. 

To further understand their similarity, we selected major marker genes from previous studies on the large intestine (Appendix A) [17], small intestine (Appendix A) [17], and pancreas (Appendix A) [18] of *M. musculus* to identify the homologous genes in *C. savignyi* intestines. The expression profiles of these genes in different intestinal segments of *C. savignyi* are displayed by heat map (Figure 4C–E). The results showed that the homologous genes of the small intestine in vertebrates were highly expressed in the digestion and absorption segments. The homologous genes of the large intestine have a high expression in the immune segment. The pancreatic homologs of the vertebrates (*M. musculus*) were expressed in the *C. savignyi* digestion segment. Many pancreatic homologs were also highly expressed in the stomach (Appendix A), consistent with a previous study in *C. robusta* [15,19]. To corroborate marker gene expression on the different intestinal segments in *C. savignyi*, we performed *Aldh1a2* (small intestine marker gene), *lgfbp5* (large intestine marker gene) in vitro RNA quantification experiments. qPCR results showed that the relative expression level of *Aldh1a2* in the digestion segment was significantly higher than that in the absorption and immune segments (Figure 4F). The relative expression level of *lgfbp5* gene in the immune segment was significantly higher than that in the digestion and digestion segments (Figure 4G). In *C. savignyi* intestine data, a total of 12 *trypsin*-related genes were identified in all intestinal segments with high expression in the stomach and digestion segments (Figure 4H). The relative expression level of *TRYPSIN type protease2* (ENSAVG00000008397) gene in the digestion segment was significantly higher than that in the absorption and immune segments (Figure 4I).

Taken together, the digestion and absorption segments and the immune segment of *C. savignyi* have a similar function to the small intestine and the large intestine of the vertebrates, respectively. Meanwhile, pancreatic homologs of the vertebrates (*M. musculus* and *H. sapiens*) were expressed in the *C. savignyi* digestion segment, implying that the digestive segment plays the part of the pancreas’ functions in *C. savignyi*. 

### 2.4. The Multi−Species Comparison Explains the Conservation and Divergence of Intestine Function during Evolution

To explore the conservation and variability of the intestine during evolution, comparative analyses of multi-species sequences were carried out on the intestines of *A. japonicus, D. rerio, X. laevis, H. sapiens*, *C. robusta*, and *C. savignyi*. A genome-wide phylogenetic tree was constructed based on single-copy genes from genomes of the six species mentioned above (Figure 5A). The results showed that 511 families were expanded and 1640 gene families were contracted in the intestine of *C. savignyi*. The expanded gene families including the BACK, BTB, Histone, Fibrinogen_C, PrkA, and Glyco_transf_10 were rapidly evolving (Appendix A). GO enrichment analysis showed that the expanded families’ genes were enriched in glycosylation, metabolic process, and symporter activity (Appendix A). 

Next, we conducted an analysis of the intestine shared conserved orthologue genes across six species. The numbers of the shared orthologous genes in six species are 7101 (*C. robusta*), 2526 (*A. japonicus*), 3524 (*D. rerio*), 2557 (*X. laevis*), and 2136 (*H. sapiens*), respectively (Figure 5B). *C. savignyi* shared the most orthologous genes with the same genus, *C. robusta*. GO enrichment analysis demonstrated that the orthologous genes were highly enriched in nuclear lumen, Ras protein signal transduction, lipid binding, enzyme binding, zinc ion binding, nucleus, Ras GTPase binding, small GTPase, and GTPase mediated signal/binding in invertebrates. In vertebrates, the orthologous genes were highly enriched in endoplasmic reculum subcompartment, mitochondrial protein complex, and guany nucleotide binding (Figure 5C). Six species shared 67 single-copy genes, among which some genes (*RPL18A, GOT1, PPIL1, SMAD4, IMP3, RRAGA, CPSF5, OXA1I, CCDC6, BCAS2, PGLS,* and *PSMD7*) (Figure 5D and Appendix A) are highly expressed in *H. sapiens* and enriched in structural molecule activity, cytoplasm, ribosome, macromolecule localization, and metabolic/biosynthetic process (Appendix A). Single-copy genes were enriched in the KEGG pathway of ribosome (Appendix A).

## 3. Discussion

In recent years, the intestine has become an important model for the study of fat storage disorders, diabetes, and stem cell regeneration [20,21]. In vertebrates, the intestine was subdivided into a variety of segments that play their own functions. In other model organisms with segmented intestines (for example, in the insect foregut), extraoral digestion and enzymatic conversion have been reported previously, but in *Drosophila,* it is generally accepted that the midgut is the main digestive site [4,22,23]. The *C. elegans* digestive tract is generally divided into the buccal cavity, foregut or pharynx, midgut or intestine, and hindgut, of which the intestine is responsible for digestion and absorption [6,24,25,26]. The zebrafish gut is anatomically similar to the vertebrates’, where an enlarged anterior segment is mainly used for food storage and digestion [27,28]. The midgut and hindgut are similar to those of humans and mice, which are responsible for food digestion and nutrient absorption [29,30]. 

Nine Hox genes were identified in *C. robusta* and showed a regional expression pattern in the digestive system [13,31,32]. In this study, seven Hox genes were identified in intestine transcriptome data in *C. savignyi*. They expressed in different segments in the intestine. The GO enrichment results of all genes suggest the functional differentiation of *C. savignyi* intestine. The enrichment analyses of genes in intestine segments indicate that the proximal and middle segments probably possess a potential digestion and absorption function, respectively. Furthermore, expression tendency analysis showed that the digestive function was gradually weakened along the proximal, middle, and distal segments, whereas the immune function was gradually improved. 

Functional analysis of DEGs here argues that the *C. savignyi* intestine was functionally segmented. The DEGs of the digestive segment were enriched in a large number of enzymatic metabolic pathways, including peptide, aminoglycan metabolic process, and other digestive and metabolic processes. This is consistent with the studies in vertebrates [33,34], of which the anterior intestine is the major site for digestion. Some DEGs, such as *PNLIP* and *acp5*, which are related to digestive function [35], were up-regulated in the proximal intestine. In addition to the fundamental digestive and absorption functions, the intestine is one of the largest organs of the immune system and plays a crucial role in the immunity function [5,36,37]. TNF has been identified as a significant immune function as a pathological component of autoimmune diseases [38,39]; Rho GTPase signaling pathways are a cause of many immune system-related diseases [40,41]; and Glycosylation can modulate inflammatory responses, enable viral immune escape, promote cancer cell metastasis or regulate apoptosis [42]. We found that the DEGs of the distal intestine were enriched in some immune-related pathways, including TNF receptor superfamily binding, TNF receptor binding, and Rho GTPase binding. Furthermore, the tissue-specific-expressed genes of the distal segment were related to glycosylation and glycoprotein metabolic/biosynthetic process terms. The above results indicated the potentially dominant immunity function of this segment. 

In vertebrates, the stomach is the part of the digestive system connected to the intestine, and its main function is to store and help digest food. Instead of a stomach, zebrafish have the intestinal bulb, which has a bigger lumen than the posterior part and thus may function as a reservoir comparable to the stomach [29,30]. When food enters stomach, the stomach mixes the food and liquid with digestive juices, and then slowly empties its contents into the intestine. The anterior intestines then further mix food with digestive juices and push the mixture forward for further digestion and absorption by the intestinal walls [43]. In this study, we found that the stomach functionally resembles the proximal digestion segment and the middle absorption segment of the intestines, rather than the distal immune segment. The results indicate that the tunicate *C. savignyi* shares a functional relationship with vertebrates between the intestine and stomach related to aspects of food digestion and absorption. 

The large intestine and small intestine of vertebrates have distinct functions [20]. When food moves through the intestine, epithelial cells of different intestinal segments specifically absorb components of digested food by specific transporters. For example, arginine and tryptophan are mainly absorbed by the small intestine, whereas nucleosides, organic acids, fat, and choline are absorbed by the large intestine [44,45]. The digestive tract itself has certain immune functions, but the immune functions of different intestinal segments are completely different [6]. For example, in vertebrates, the large intestine is responsible for dealing with the toxic substances in feces that can elicit an immune response against adverse effects on the function of both local and systemic immune systems, and thus has a stronger immune function. The small intestine, on the other hand, mainly absorbs various nutrients and causes little immune response, so its immune function is relatively weaker [43,46,47]. Our results showed that *C. savignyi* proximal and middle intestines were enriched with a large number of peptide metabolic/catabolic process, amide biosynthetic process, and protein serine/threonine kinase activity pathway (Appendix A and Figure 2D,E). Expression profiles of the orthologous genes in different segments by comparing marker genes of humans and mice with *C. savignyi* revealed that the digestive and absorption segments functioned similarly to the small intestine in vertebrates. The immune segment of *C. savignyi* was enriched in the pathways related to autophagy metabolic pathways and immune system process. The marker genes in the large intestine of vertebrates showed that they were highly expressed in the distal intestine of *C. savignyi*, which fully suggested that the distal intestine of *C. savignyi* was functionally similar to the large intestine of vertebrates. 

As a closest relative to vertebrates, *C. savignyi* has a relatively simple digestive system. It is known that in mice, the pancreas derives from the foregut endoderm, which is formed by a small field of epithelial precursor cells expressed by pancreatic duodenal transcription factor 1 [18]. A hepatopancreas (also called a pyloric gland), as an accessory organ, is a branching structure developed from the boundary between the stomach and intestine in ascidians [48,49]. In a previous study, the expression of *Ciona* homologs of pancreatic- and gastric-related exocrine enzyme genes and their transcriptional regulator genes is restricted to the *Ciona* stomach [15,19]. Our results showed that pancreatic marker genes have a higher expression in the stomach and proximal segment than the middle and distal intestine of *C. savignyi.* Many *trypsin* genes, including pancreatic proteolytic enzyme [50], were also found here to be expressed at the higher level in the stomach and the proximal segment of *C. savignyi* intestines. Therefore, we speculate that the proximal segment of the *C. savignyi* intestine plays part of the pancreatic function. 

In conclusion, our results demonstrated a regional variation in intestinal function of *C. savignyi*, with a progressive decrease in digestive function and a progressive increase in immune function from the proximal to the distal segments. The functions of the intestine were conserved in different species. It is suggested that the proximal and middle segments of *C. savignyi* intestines may have similar functions to the small intestine of vertebrates. The distal intestinal segment is functionally similar to the large intestine of vertebrates. Furthermore, tunicate *C. savignyi* shares the same functional relationship with vertebrates between the intestine and stomach. Our study provides a new basis for studying the intestine functions of ascidians at the molecular level, and it also offers insights into the functional evolution of the digestive system in chordates.

## 4. Materials and Methods

### 4.1. Animals and Sample Preparation 

The adults of *C. savignyi* were collected from Weihai City, China, and cultured in seawater at 18 °C in the laboratory. We selected healthy adult *C. savignyi* and starved them 3–4 days prior to dissection to collect the empty stomach and intestines using sterilized dissection tools. To facilitate sample collection, the *C. savignyi* intestine was longitudinally cut into three segments equally (Figure 1A). Each sample had three biological replicates. 

### 4.2. RNA Extraction and Transcriptome Sequencing

The total RNA of *C. savignyi* adult intestines and stomach were extracted, respectively, using RNAiso plus reagent (Takara Biomedical Technology (Beijing) Co., Ltd., Beijing, China). The integrity and quality of total RNA was determined by 1% agarose gel electrophoresis. RNA purity was checked using nanodrop spectrophotometry (Eppendorf). RNA concentration was measured using Qubit. Agilent bioanalyzer 2100 system (Agilent Technologies, Santa Clara, CA, USA) was used to assess the quality of the extracted RNA. Only high-quality RNA samples (OD260/280 = 1.96–2.06, OD260/230 ≥ 1.9, RIN ≥ 7.5, 28/18S ≥ 1.1, >35 μg) were used to construct cDNA libraries. 

The libraries were constructed following the NEB common library construction method [51]. Briefly, mRNA was purified from total RNA using poly-T oligo attached magnetic beads. The first strand of cDNA was synthesized using random primers, and the second strand of cDNA synthesis was subsequently performed using buffer, dNTPs, DNA polymerase I, and RNase H. The library fragments were purified with beads, then terminal repair, adapter, and A-tailing were added. RNA-seq libraries (Illumina) with insert sizes of 200–300 bp cDNA were prepared. The target length fragments were captured and subjected to PCR amplification to complete the library construction. Lastly, the libraries were sequenced on Illumina novaseq using paired−end 150 base pair reading at Novogene Technologies (Tianjin, China).

### 4.3. Transcriptome Analysis

At least 1.5 million reads were obtained from each sample. The clean reads were obtained for subsequent analysis after filtration of original data. The filtered reads with N bases of more than 10%, and reads with low−quality bases (≤5) of more than 50%, were removed. The clean reads were then compared to the reference genome (*C. savignyi*, CSAV2.0; Ensembl, https://asia.ensembl.org/index.html, accessed on 2 July 2021) by use of STAR software [52]. By using SMRT-seq mapped clean data to reference sequences, reads count value of genes was obtained [53]. Gene expression levels were estimated by HTSeq [54], and all counts were converted to FPKM value.

Differential expression analysis of two conditions/groups was performed using the DESeq2 R package (1.18.0) according to the manual [55]. The resulting *p*-values were adjusted using Benjamini and Hochberg’s approach for controlling the false discovery rate (FDR < 0.05). Genes with adjusted *p*-values < 0.05 were considered to be differentially expressed. GO and KEGG pathway enrichment of genes was analyzed using clusterProfiler package [56,57,58,59].

### 4.4. Combinatorial Analyses of Transcriptome Data

The transcript data from each of the three intestinal segments were compared with those from the stomach, focusing on the different function between the proximal intestine and stomach. Screening of differentially expressed genes (DEGs) was performed as described above in three segments of the intestine as well as the stomach. The relationship between the intestine and the stomach was analyzed based on the Pearson correlation data [60].

### 4.5. Expression Tendency Analysis of all Genes 

The tendency expression analysis of all genes in the intestine was conducted to identify the cluster with biologically similar functions. The analysis method was performed using the OmicShare tools, an online platform for data analysis (http://www.omicshare.com/tools, accessed on 10 September 2022).

### 4.6. Selection and Phylogenetic Analysis of Orthologous Genes

The gene families were determined via an all-to-all blastp search against the intestine protein-coding genes from the selected deuterostomes including human (*H. sapiens*), frog (*Xenopus laevis*), zebrafish (*Danio rerio*), and sea cucumber (*Apostichopus japonicus*). Orthologous relationships between genes were inferred through all-against-all protein sequence similarity searches using OrthoFinder 2.2 [61]. Using OthoMCL (http://orthomcl.org/orthomcl/, 10 September 2022), we inferred orthologous relationships between genes, and retained only the longest predicted transcript. The phylogenetic tree was constructed based on one-to-one orthologs, which were clustered among protein-coding genes of the selected five genomes. The divergence time was estimated by mcmctree (http://abacus.gene.ucl.ac.uk/software/paml.html, 10 September 2022) in PAML (version 4.5) [62]. The expansion and contraction gene families were estimated using CAFE (version 5.0) [63,64]. 

### 4.7. The Alignment Analysis of Homologous Genes of C. savignyi against Multiple Species 

*C. robusta* transcriptome data were downloaded from the NCBI database (PRJNA731286) [16]. *C. robusta* proteome data were downloaded from Ensembl v96 (http://ghost.zool.kyotou.ac.jp/default_ht.html, 10 September 2022) [65]. The amino acid sequences corresponding to the genes expressed in each segment of the intestine were extracted, and the data from *M. musculus* and *H. sapiens* were subjected to two−way BLAST to search for the optimal result. The threshold was set at e−value of <1 × 10^−5^. The resultant BLASTP hits were considered as *C. robusta* genes homologous to *C. savignyi,* while the genes lacking hits in BLASTP were considered as *C. savignyi*-specific.

Transcriptome and proteome data of other organisms, including *A. japonicus* (SRX8950439) [66], *D. rerio* (SRX10579902, SRX10579903, SRX10579904) [67], *X. laevis* (SRX1286475) [68], and *H. sapiens* (ERX288571, ERX288581, ERX288502) [69] were downloaded from the EDomics website (http://edomics.qnlm.ac, 10 September 2022) [70]. Sratools (https://hpc.nih.gov/apps/sratoolkit.html, 10 September 2022) software was used to process the downloaded SAR data to obtain FASTQ files and corresponding Sam files. STAR2.0.2 (https://github.com/alexdobin/STAR, 10 September 2022) [52] was used to compare the raw data, and feature counts (http://subread.sourceforge.net/, 10 September 2022) were used to quantitatively calculate the counts value of each gene to finally obtain the corresponding TPM value. The value of TPM > 1 was used to screen genes expressed in the intestine of the corresponding species, and the corresponding amino acid sequences of the genes were selected for multi−species alignment. The threshold was set at e-value < 1 × 10^−5^. According to the expression of homologous genes of each species in *C. savignyi*, GO and KEGG enrichment analyses were performed.

### 4.8. Quantitative Reverse−Transcription PCR (qRT−PCR)

Reliability verification of the transcriptome results was performed by qRT−PCR, which also was used for expression analysis of genes. The first−strand cDNA was synthesized using 1 μg total RNA per 20 μL reaction system by reverse transcriptase (Vazyme). The primers of RT-qPCR were designed in the website PrimerBank (https://pga.mgh.harvard.edu/primerbank/index.html, 10 September 2022). RT-qPCR was performed using the ChamQ SYBR Color qPCR Master Mix (Vazyme Biotech Co., Ltd., Nanjing, China) on Light Cycler 480 (Roche). All RT-qPCR primers are listed in Appendix A. EF1α was used as the reference gene. Data were calculated using the 2^−ΔΔCt^ method.

### 4.9. Data Analysis

If not specified, statistics and visualization were performed in R studio (https://www.rstudio.com/, 10 September 2022), and figure formats were unified in Adobe Illustrator CC 2018 to modify typesetting. All data statistical analysis was performed using paired Student’s *t*-tests. The *p*-value < 0.05 was considered to be a significant difference. * represents 0.01 < *p* < 0.05. ** represents 0.001 < *p* < 0.01. *** represents *p* < 0.001.

## Figures and Tables

**Figure 1 ijms-24-06270-f001:**
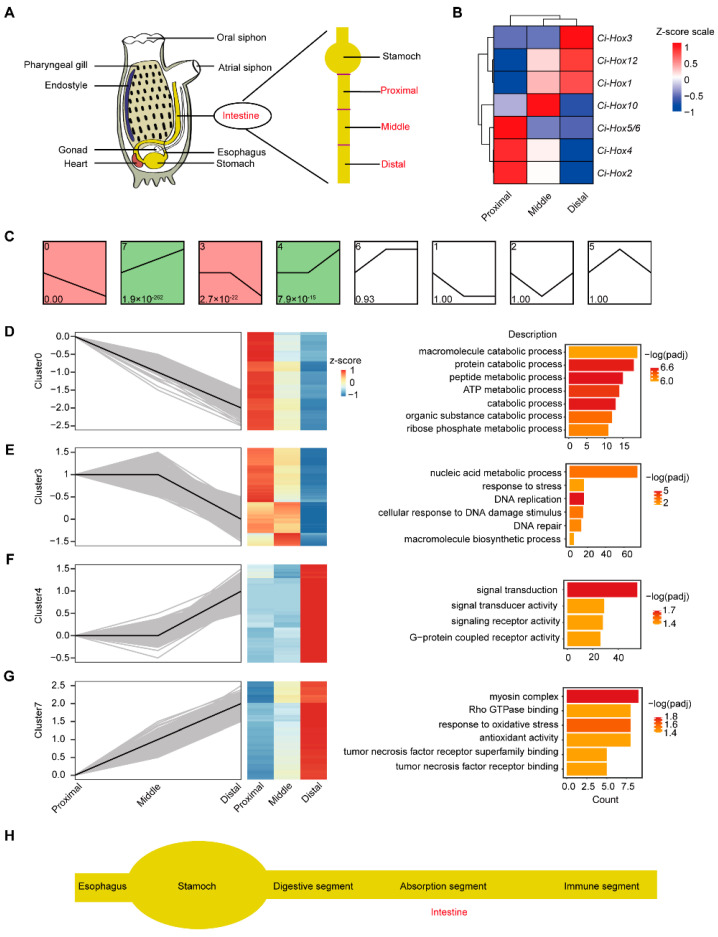
Intestinal function of different segments. (**A**) *Ciona* anatomical diagram. Left is the diagram illustrating adult organs and tissues. Yellow color indicates stomach and intestine. Right is the diagram of intestinal segments. (**B**) Heat map displays Z-score transformed expression of 7 Hox genes specific to the proximal, middle, and distal segments. (**C**) Expression modules were identified using expression tendency analysis in digestive, absorption, and immune segments. A total of eight clusters were divided. The colored modules are the modules with significant differences. The cluster IDs and *p*-values are shown at the upper left and lower left of the boxes, respectively. The horizontal line represents the expression trend in different samples. (**D**–**G**) The gene expression trend of clusters 0, 3, 4, and 7 of the digestive, absorption, and immune segments. The transcript levels of genes were quantified by row Z-score and represented as heat maps. The right was GO enrichment analysis for corresponding cluster genes. (**H**) The schematic illustration of different intestine segments.

**Figure 2 ijms-24-06270-f002:**
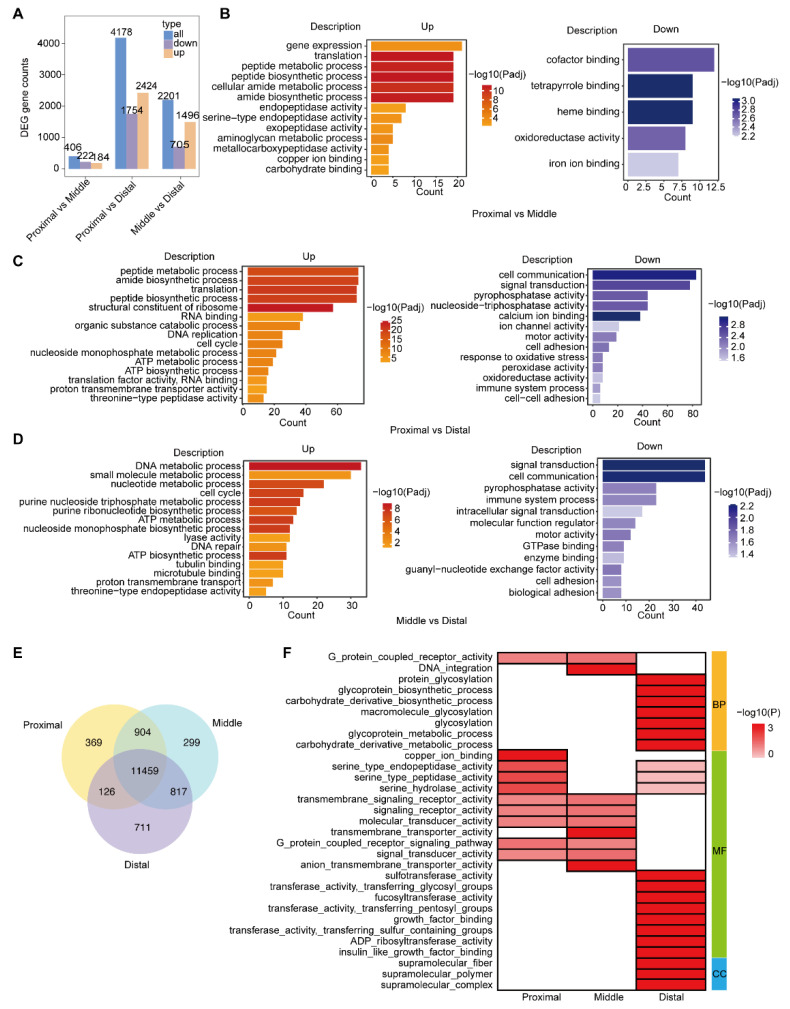
DEGs and specific genes enrichment analysis in different intestine segments. (**A**) The number of differential expression genes. (**B**–**D**) GO enrichment analyses for up-regulated genes and down-regulated genes. (**E**) Venn diagram of common and unique genes among the three segments. (**F**) Heat map diagram of GO enrichment of specific genes in the three segments. Color indicates −log(P), dark indicates high enrichment.

**Figure 3 ijms-24-06270-f003:**
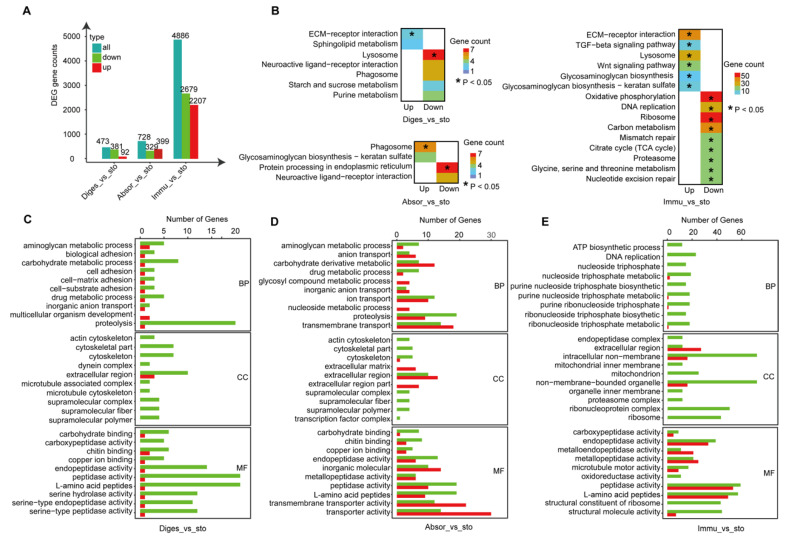
Comparison of the digestive segment, middle, and distal intestine with the stomach. (**A**) DEGs between stomach and digestive segment, absorption segment, and immune segment. (**B**) KEGG pathway analysis for up-regulated genes and down-regulated genes in digestive segment vs. sto, absorption segment vs. sto, and immune segment vs. sto. (**C**–**E**) GO enrichment analysis for up-regulated genes and down-regulated genes in digestive segment vs. sto, absorption segment vs. sto, and immune segment vs. sto. The colors correspond to the A panel. (Sto: stomach, Diges: digestive segment, Absor: absorption segment, Immu: immune segment).

**Figure 4 ijms-24-06270-f004:**
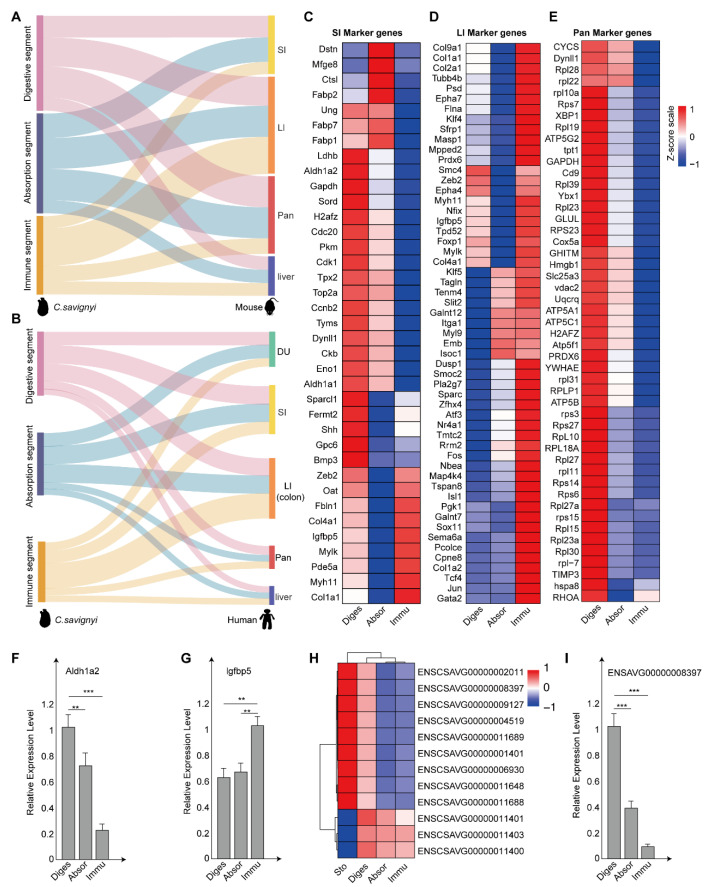
Comparison of the *C. savignyi* intestine with the digestive system of humans (*Homo sapiens*) and mice (*Mus musculus*). (**A**,**B**) Sankey plot showing the connectivity between the intestine of *C. savignyi* and the digestive system of *Mus musculus* (**A**) and *H. sapiens* (**B**). The thickness of the line represents the number of orthologous genes. (**C**–**E**) Heat map of expression of homologues of *Mus musculus* SI, LI, and Pan marker genes in *C. savignyi* stomach and different intestinal segments. The row Z-score was calculated by log_2_-transformed FPKM values of targeted genes. The formula to calculate row Z-scores from FPKM values was log_2_(FPKM + 1). The row Z-score value ranges from −1 to 1 and is represented as a color-coded box, with red and blue indicating relative up−regulation and down−regulation, respectively. (**F**,**G**,**I**) qPCR confirms mRNA level of *Aldhla2* (SI marker gene), *lgfbp5* (LI marker genes), and *TRYPSIN* (Pan marker genes) (**H**) Gene expression heat map of pan markers genes in Sto, Diges, Absor, and Immu. Error bars represent standard deviation. *** *p* < 0.001 and ** *p* < 0.01 by Student’s *t*-test. (Sto: stomach, Diges: digestive segment, Absor: absorption segment, Immu: immune segment, DU: duodenum, SI: small intestine, LI: large intestine, Pan: pancreas).

**Figure 5 ijms-24-06270-f005:**
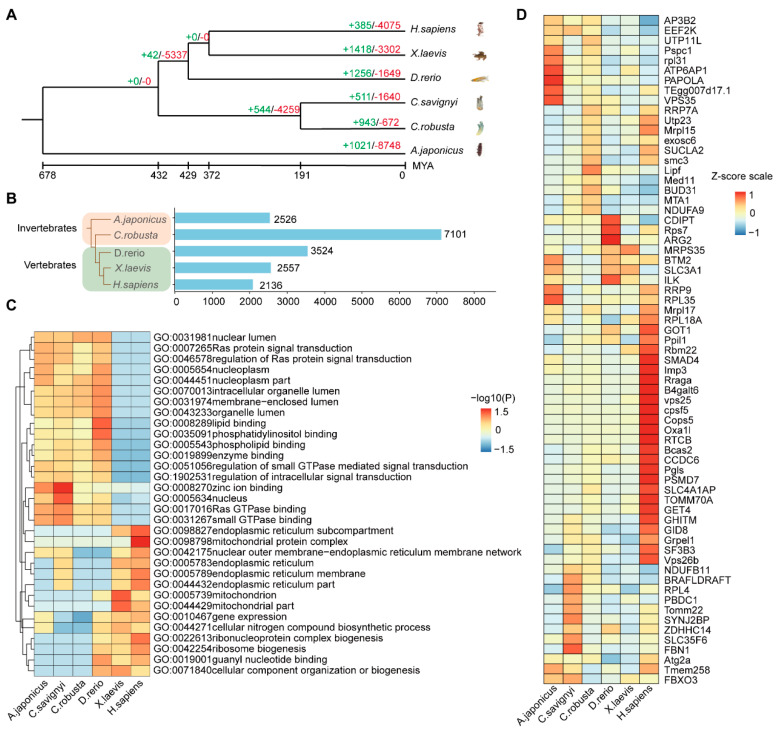
Conservation and divergence of intestine function during evolution. (**A**) The phylogenetic position, divergence time estimation, and gene family analysis of *C. savignyi* intestine. The green and red numbers indicate the expanded gene families and the contracted gene families, respectively. The divergence years are displayed below the phylogenetic tree. (**B**) Bar chart showing the numbers of intestine orthologous genes in different organisms compared with *C. savignyi*. (**C**) GO enrichment analysis of the orthologous genes cross-species similarities; *p*-value was derived by a hypergeometric test. (**D**) The heat map of single-copy orthologous genes expression in cross-species.

## Data Availability

The raw data can be found in online repositories.

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
