# Peer review of "Comparative Transcriptomic Analysis Reveals the Functionally Segmented Intestine in Tunicate Ascidian"

_ijms, 2023, doi:10.3390/ijms24076270_

Round 1

Reviewer 1 Report

Except for a few points, it is a well-prepared manuscript.

- Abstract: there is a gap between Line 17-18

- Between Line 60-70: It is a conclusion sentence, and should not be given at the end of the introduction. This part needs to be rewritten. 

- Line 439-440: ‘The total RNA of C. savignyi adult intestines and stomach were extracted using RNAiso plus reagent (Takara), respectively.

‘Respectively’ represents what?

 - Line 478-481: ‘The tendency expression analysis of intestine-all genes was conducted to identify the cluster with biologically similar functions. The analysis method was performed using the OmicShare tools, a free online platform for data analysis (http://www.om icshare.com/tools).’

It does not look like a free for all analysis.

Author Response

Q1: Abstract: there is a gap between Line 17-18

ResponseWe don’t know what happened when the manuscript was submitted. We now have removed the gap.

Q2: Between Line 60-70: It is a conclusion sentence, and should not be given at the end of the introduction. This part needs to be rewritten. 

ResponseThanks for the comments. We have modified this paragraph as “In the present study, we carried out the transcriptome sequencing of the whole intestine of ascidian C. savignyi to build and analyze its expression profile. Then, comparative transcriptome analyses was performed to find out the functional relationship between intestine and stomach of C. savignyi. Subsequently, expressions of the orthologs of vertebrate digestive organ-specific genes in the of C. savignyi intestine were analyzed to search the connectivity between the intestine of C. savignyi and the digestive system of vertebrates. Ultimately, we also did the multi-species comparison on the conservation and divergence of intestine function during evolution.”

Q3: Line 439-440: ‘The total RNA of C. savignyi adult intestines and stomach were extracted using RNAiso plus reagent (Takara), respectively.’ ‘Respectively’ represents what?

Response: We are sorry for the confusion, and modified as “The total RNA of C. savignyi adult intestines and stomach were extracted, respectively, using RNAiso plus reagent (Takara)”.

Q4: Line 478-481: ‘The tendency expression analysis of intestine-all genes was conducted to identify the cluster with biologically similar functions. The analysis method was performed using the OmicShare tools, a free online platform for data analysis (http://www.omicshare.com/tools).’ It does not look like a free for all analysis.

Response: We are sorry for the mistake, and have deleted the “free”.

Reviewer 2 Report

In the manuscript, Zhang et al. investigated the segmentation of the intestine in Ciona savignyi – a tunicate ascidian species. They showed three functionally distinct segments in the intestine of Ciona savignyi by performing RNA sequencing and studying gene expression profiles. Based on the present study, the authors have concluded that the proximal segment might have roles in digestive function, the middle segment might have roles in absorption function, and the distal segment might be important for immune function. 

Overall, the manuscript is well written, and it would be interesting for the readers. Included references are relevant. Methods are sufficiently described. The study is well organized and discussed. The presented manuscript could be considered for publication in my opinion.    

Author Response

Thanks very much for your comments on our manuscript.